# MULTI-LEVEL GENERATIVE MODELS FOR PARTIAL LABEL LEARNING WITH NON-RANDOM LABEL NOISE

## ABSTRACT

Partial label (PL) learning tackles the problem where each training instance is associated with a set of candidate labels that include both the true label and irrelevant noise labels. In this paper, we propose a novel multi-level generative model for partial label learning (MGPLL), which tackles the PL problem by learning both a label level adversarial generator and a feature level adversarial generator under a bi-directional mapping framework between the label vectors and the data samples. MGPLL uses a conditional noise label generation network to model the non-random noise labels and perform label denoising, and uses a multi-class predictor to map the training instances to the denoised label vectors, while a conditional data feature generator is used to form an inverse mapping from the denoised label vectors to data samples. Both the noise label generator and the data feature generator are learned in an adversarial manner to match the observed candidate labels and data features respectively. We conduct extensive experiments on both synthesized and real-world partial label datasets. The proposed approach demonstrates the state-of-the-art performance for partial label learning.

## 1 INTRODUCTION

Partial label (PL) learning is a weakly supervised learning problem with ambiguous labels (Hüllermeier & Beringer, 2006; Zeng et al., 2013), where each training instance is assigned a set of candidate labels, among which only one is the true label. Since it is typically difficult and costly to annotate instances precisely, the task of partial label learning naturally arises in many real-world learning scenarios, including automatic face naming (Hüllermeier & Beringer, 2006; Zeng et al., 2013), and web mining (Luo & Orabona, 2010).

As the true label information is hidden in the candidate label set, the main challenge of PL lies in identifying the ground truth labels from the candidate noise labels, aiming to learn a good prediction model. Some previous works have made effort on adjusting the existing effective learning techniques to directly handle the candidate label sets and perform label disambiguation implicitly (Gong et al., 2018; Nguyen & Caruana, 2008; Wu & Zhang, 2018). These methods are good at exploiting the strengths of the standard classification techniques and have produced promising results on PL learning. Another set of works pursue explicit label disambiguation by trying to identify the true labels from the noise labels in the candidate label sets. For example, the work in (Feng & An, 2018) tries to estimate the latent label distribution with iterative label propagations and then induce a prediction model by fitting the learned latent label distribution. Another work in (Lei & An, 2019) exploits a self-training strategy to induce label confidence values and learn classifiers in an alternative manner by minimizing the squared loss between the model predictions and the learned label confidence matrix. However, these methods suffer from the cumulative errors induced in either the separate label distribution estimation steps or the error-prone label confidence estimation process. Moreover, all these methods have a common drawback: they automatically assumed random noise in the label space – that is, they assume the noise labels are randomly distributed in the label space for each instance. However, in real world problems the appearance of noise labels is usually dependent on the target true label. For example, when the object contained in an image is a "computer", a noise label "TV" could be added due to a recognition mistake or image ambiguity, but it is less likely to annotate the object as "lamp" or "curtain", while the probability of getting noise labels such as "tree" or "bike" is even smaller.

In this paper, we propose a novel multi-level adversarial generative model, MGPLL, for partial label learning. The MGPLL model comprises of conditional data generators at both the label level and feature level. The noise label generator directly models non-random appearances of noise labels conditioning on the true label by adversarially matching the candidate label observations, while the data feature generator models the data samples conditioning on the corresponding true labels by adversarially matching the observed data sample distribution. Moreover, a prediction network is incorporated to predict the denoised true label of each instance from its input features, which forms inverse mappings between labels and features, together with the data feature generator. The learning of the overall model corresponds to a minimax adversarial game, which simultaneously identifies true labels of the training instances from both the observed data features and the observed candidate labels, while inducing accurate prediction networks that map input feature vectors to (denoised) true label vectors. To the best of our knowledge, this is the first work that exploits multi-level generative models to model non-random noise labels for partial label learning. We conduct extensive experiments on real-world and synthesized PL datasets. The empirical results show the proposed MGPLL achieves the state-of-the-art PL performance.

## 2 RELATED WORK

Partial label (PL) learning is a popular weakly supervised learning framework (Zhou, 2018) in many real-world domains, where the true label of each training instance is hidden within a given candidate label set. The challenge of PL learning lies in disambiguating the true labels from the candidate label sets to induce good prediction models.

One strategy towards PL learning is to adjust the standard learning techniques and implicitly disambiguate the noise candidate labels through the statistical prediction pattern of the data. For example, with the maximum likelihood techniques, the likelihood of each PL training sample can be defined over its candidate label set instead of its implicit ground-truth label (Jin & Ghahramani, 2003; Liu & Dietterich, 2012). For the $k$-nearest neighbor technique, the candidate labels from neighbor instances can be aggregated to induce the final prediction on a test instance (Hüllermeier & Beringer, 2006; Gong et al., 2018; Zhang & Yu, 2015). For the maximum margin technique, the classification margin can be defined over the predictive difference between the candidate labels and the non-candidate labels for each PL training sample (Nguyen & Caruana, 2008; Yu & Zhang, 2016). For the boosting technique, the weight of each PL training instance and the confidence value of each candidate label being ground-truth label can be refined via each boosting round (Tang & Zhang, 2017). For the error-correcting output codes (ECOC) technique, multiple binary classifier corresponding to the ECOC coding matrix are built based on the transformed binary training sets (Zhang et al., 2017). For the binary decomposition techniques, a one-vs-one decomposition strategy has been adopted to address PL learning by considering the relevance of each label pair (Wu & Zhang, 2018).

Recently, there have been increasing attentions in designing explicit feature-aware disambiguation strategies (Feng & An, 2018; Xu et al., 2019a; Feng & An, 2019; Wang et al., 2019a). The authors of (Feng & An, 2018) attempt to refine the latent label distribution using iterative label propagations and then induce a predictive model based on the learned latent label distribution. However, the latent label distribution estimation in this approach can be impaired by the cumulative error induced in the propagation process, which can consequently degrade the PL learning performance, especially when the noisy labels dominate. Another work in (Lei & An, 2019) tries to refine the label confidence values with a self-training strategy and induce the prediction model over the refined label confidence scores via alternative optimization. Its estimation error on confidence values however can negatively impact the coupled partial label classifier due to the nature of alternative optimization. A recent work in (Yao et al., 2020) proposes to address the PL learning problem by enhancing the representation ability via deep features and improving the discrimination ability through margin maximization between the candidate labels and the non-candidate labels. Another recent work in (Yan & Guo, 2020) proposes to dynamically correct label confidence values with a batch-wise label correction strategy and induce a robust predictive model based on the MixUp enhanced data. Although these works demonstrate good empirical performance, they are subject to one common drawback of assuming random distributions of noise labels by default, which does not hold in many real-world learning scenarios. This paper presents the first work that explicitly model non-random noise labels for partial label learning.

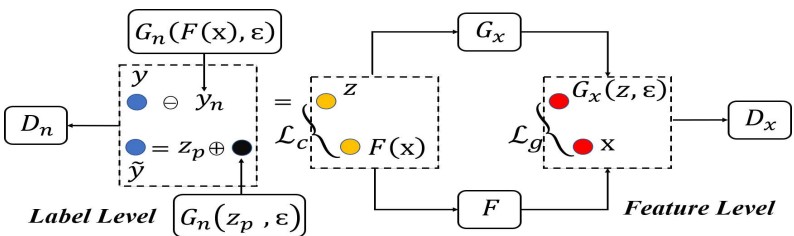

Figure 1: The proposed MGPLL model. It consists of an adversarial generative model at the label level with a conditional noise label generator $G_n$ and a discriminator $D_n$, and an adversarial generative model at the feature level with a conditional sample generator $G_x$ and a discriminator $D_x$. The prediction network $F$ builds connections between these two level generative models while providing an inverse mapping for $G_x$.

PL learning is related to other types of weakly supervised learning problems, including noise label learning (NLL) (Xu et al., 2019b; Thekumparampil et al., 2018; Arazo et al., 2019) and partial multi-label learning (PML) (Wang et al., 2019b; Fang & Zhang, 2019; Xie & Huang, 2018), but addresses different problems from them. The main difference between the PL learning and the other two well-established learning problems lies in the assumption on the label information provided by the training samples. Both PL learning and NLL aim to induce a multi-class prediction model from the training instances with noise-corrupted labels. However NLL assumes the true labels on some training instances are replaced by the noise labels, while PL assumes the true-label coexists with the noise labels in the candidate label set of each training instance. Hence the off-the-shelf NLL learning methods cannot be directly applied to solve the PL learning problem. Both PL learning and PML learn from training samples with ambiguous candidate label sets, which contains the true labels and additional noise labels. But PL learning addresses a multi-class learning problem where each candidate label set contains only one true label, while PML learning addresses a multi-label learning problem where each candidate label set contains all but unknown number of true labels.

The Wasserstein Generative Adversarial Networks (WGANs) (Arjovsky et al., 2017), which perform minimax adversarial training with a generator and a discriminator, is a popular alternative to the standard GANs (Goodfellow et al., 2014b) due to its effective and stable training of GANs. During the past few years, WGANs have been proposed as a successful tool for various applications, including adversarial sample generation (Zhao et al., 2017), domain adaption (Dou et al., 2018), and learning with noisy labels (Chen et al., 2018). This paper presents the first work that exploits WGAN to model non-random noise labels for partial label learning.

## 3 PROPOSED APPROACH

Given a partial label training set $S = \{(\mathbf{x}_i, \mathbf{y}_i)\}_{i=1}^{n}$, where $\mathbf{x}_i \in \mathbb{R}^d$ is a $d$-dimensional feature vector for the $i$-th instance, and $\mathbf{y}_i \in \{0, 1\}^L$ denotes the candidate label indicator vector associated with $\mathbf{x}_i$, which has multiple 1 values corresponding to the ground-truth label and the additional noise labels, the task of PL learning is to learn a good multi-class prediction model from $S$. In real world scenarios, the irrelevant noise labels are typically not presented in a random manner, but rather correlated with the ground-truth label. In this section, we present a novel multi-level generative model for partial label learning, MGPLL, which models non-random noise labels using an adversarial conditional noise label generator, and builds connections between the denoised label vectors and instance features using a label-conditioned feature generator and a label prediction network. The overall model learning problem corresponds to a minimax adversarial game, which conducts multi-level generator learning by matching the observed data in both the feature and label spaces, while boosting the correspondence relationships between features and labels to induce an accurate multi-class prediction model.

Figure 1 illustrates the proposed multi-level generative model, MGPLL, which attempts to address the partial label learning problem from both the label level and feature level under a bi-directional mapping framework. The MGPLL model comprises five component networks: the conditional noise

label generator, $G_n$, which models the noise labels conditioning on the ground-truth label at the label level; the conditional data generator, $G_x$, which generates data samples at the feature level conditioning on the denoised label vectors; the discriminator, $D_n$, which separates the generated candidate label vectors from the observed candidate label vectors in the real training data; the discriminator, $D_x$, which separates the generated samples from the real data in the feature space; and the prediction network, $F$, which predicts the denoised label for each sample from its input features. $\mathbf{z}_p$ denotes a one-hot label indicator vector sampled from a multinomial distribution $P_{\mathbf{z}}$. The conditional noise label generator $G_n$ induces the denoised prediction target for the prediction network $F$, while the conditional data generator $G_x$ learns an inverse mapping at the feature level that maps the denoised label vectors in the label space to the data samples in the feature space. Below we present the details of the two level generations and the overall learning algorithm.

## 3.1 CONDITIONAL NOISE LABEL GENERATION

The key challenge of partial label learning lies in the fact that the ground-truth label is hidden among the noise labels in the given candidate label set. As aforementioned, in real world partial label learning problems, the presence of noise labels typically does not happen at random, but rather correlates with the ground-truth labels. Hence we propose a conditional noise label generation model to model the appearances of the target-label dependent noise labels by adversarially matching the observed candidate label distribution in the training data, aiming to help identify the true labels later.

Specifically, given a noise value sampled from a uniform distribution $\epsilon \sim P_\epsilon$ and a one-hot label indicator vector $\mathbf{z}$ sampled from a multinomial distribution $P_{\mathbf{z}}$, we use a noise label generator $G_n(\mathbf{z}, \epsilon)$ to generate a noise label vector conditioning on the true label $\mathbf{z}$, which can be combined with $\mathbf{z}$ in a rectified sum, "$\oplus$", to form a generated candidate label vector $\tilde{\mathbf{y}}$, such that

$$\tilde{\mathbf{y}} = G_n(\mathbf{z}, \epsilon) \oplus \mathbf{z} = \min(G_n(\mathbf{z}, \epsilon) + \mathbf{z}, 1). \tag{1}$$

Here we assume the generator $G_n$ generates non-negative values. We then adopt the adversarial learning principle to learn such a noise label generation model by introducing a discriminator $D_n(\mathbf{y})$, which is a two-class classifier and predicts how likely a given label vector $\mathbf{y}$ comes from the real data instead of the generated data. By adopting the adversarial loss of the Wasserstein Generative Adversarial Network (WGAN), our adversarial learning problem can be formulated as the following minimax optimization problem:

$$\min_{G_n} \max_{D_n} \quad \mathcal{L}_{adv}^n(G_n, D_n) = \mathbb{E}_{(\mathbf{x}_i, \mathbf{y}_i) \sim S} D_n(\mathbf{y}_i) - \mathbb{E}_{\substack{\mathbf{z} \sim P_{\mathbf{z}} \\ \epsilon \sim P_\epsilon}} D_n(G_n(\mathbf{z}, \epsilon) \oplus \mathbf{z}) \tag{2}$$

Here the discriminator $D_n$ attempts to maximally distinguish the generated candidate label vectors from the observed candidate label indicator vectors in the real training data, while the generator $G_n$ tries to generate noise label vectors and hence candidate label vectors that are similar to the real data in order to maximally confuse the discriminator $D_n$. By playing a minimax game between the generator $G_n$ and the discriminator $D_n$, the adversarial learning is expected to induce a generator $G_n^*$ such that the generated candidate label distribution can match the observed candidate label distribution in the training data. We adopt the training loss of the WGAN here, as WGANs can overcome the mode collapse problem and have improved learning stability comparing to the standard GAN models (Arjovsky et al., 2017).

Note although the proposed generator $G_n$ is designed to model true-label dependent noise labels, it can be easily modified to model random noise label distributions by simply dropping the label vector input from the generator, which yields $G_n(\epsilon)$.

## 3.2 PREDICTION NETWORK

The ultimate goal of partial label learning is to learn an accurate prediction network $F$. To train a good predictor, we need to obtain denoised labels on the training data. For a candidate label indicator vector $\mathbf{y}$, if the noise label indicator vector $\mathbf{y}_n$ is given, one can simply perform label denoising as follows to obtain the corresponding true label vector $\mathbf{z}$:

$$\mathbf{z} = \mathbf{y} \ominus \mathbf{y}_n = \max(\mathbf{y} - \mathbf{y_n}, 0) \tag{3}$$

Here the rectified minus operator "$\ominus$" is introduced to generalize the standard minus "$-$" operator into the non-ideal case, where the noise label indicator vector $\mathbf{y}_n$ is not properly contained in the candidate label indicator vector.

The generator $G_n$ presented in the previous section provides a mechanism to generate noise labels and denoise candidate label sets, but requires the true target label vector as its input. We propose to use the outputs of the prediction network $F$ to approximate the target true label vectors of the training data for the purpose of denoising the candidate labels with $G_n$, while using the denoised labels as the prediction target for $F$. Specifically, with the noise label generator $G_n$ and the predictor $F$, we perform partial label learning by minimizing the following classification loss on the training data $S$:

$$\min_{F, G_n} \quad \mathcal{L}_c(F, G_n) = \mathbb{E}_{\substack{\epsilon \sim P_\epsilon \\ (\mathbf{x}_i, \mathbf{y}_i) \sim S}} \ell_c\big(F(\mathbf{x}_i), \ \mathbf{y}_i \ominus G_n(F(\mathbf{x}_i), \epsilon)\big) \tag{4}$$

Although in the ideal case, the output vectors of $G_n$ and $F$ would be indicator label vectors, it is error-prone and difficult for neural networks to output discrete values. To pursue more reliable predictions and avoid overconfident outputs, we use $G_n$ and $F$ to predict the probability of each class label being a noise label and the ground-truth label respectively. Hence the loss function $\ell_c(\cdot, \cdot)$ in Eq.(4) above denotes a mean square error loss between the predicted probability of each label being the true label (through $F$) and its denoised confidence of being a ground-truth label (through $G_n$).

### 3.3 CONDITIONAL FEATURE LEVEL DATA GENERATION

With the noise label generation model and the prediction network above, the observed training data in both the label and feature spaces are exploited to recognize the true labels and induce good prediction models. Next, we incorporate a conditional data generator $G_x(\mathbf{z}, \epsilon)$ at the feature level to map (denoised) label vectors in the label space into instances in the feature space, aiming to further strengthen the mapping relations between data samples and the corresponding labels, enhance label denoising and hence improve the partial label learning performance. Specifically, given a noise value $\epsilon$ sampled from a uniform distribution $P_\epsilon$ and a one-hot label vector $\mathbf{z}$ sampled from a multinomial distribution $P_\mathbf{z}$, $G_x(\mathbf{z}, \epsilon)$ generates an instance in the feature space that is corresponding to label $\mathbf{z}$. Given the training label vectors in $S$ denoised with $G_n$, the data generator $G_x$ is also expected to regenerate the corresponding training instances in the feature space. This assumption can be captured using the following generation loss:

$$\mathcal{L}_g(F, G_n, G_x) = \mathbb{E}_{\substack{(\mathbf{x}_i, \mathbf{y}_i) \sim S \\ \epsilon_1, \epsilon_2 \sim P_\epsilon}} \ell_g\big(G_x(\mathbf{z}_i, \epsilon_2), \mathbf{x}_i\big) \tag{5}$$

$$\text{with} \quad \mathbf{z}_i = \mathbf{y}_i \ominus G_n(F(\mathbf{x}_i), \epsilon_1)$$

where $\mathbf{z}_i$ denotes the denoised label vector for the $i$-th training instance, and $\ell_g(\cdot, \cdot)$ is a mean square error loss function.

Moreover, by introducing a discriminator $D_x(\mathbf{x})$, which predicts how likely a given instance $\mathbf{x}$ is real, we can deploy an adversarial learning scheme to learn the generator $G_x$ through the following minimax optimization problem with the WGAN loss:

$$\min_{G_x} \max_{D_x} \ \mathcal{L}^x_{adv}(G_x, D_x) = \mathbb{E}_{(\mathbf{x}_i, \mathbf{y}_i) \sim S} D_x(\mathbf{x}_i) - \mathbb{E}_{\substack{\mathbf{z} \sim P_\mathbf{z} \\ \epsilon \sim P_\epsilon}} D_x(G_x(\mathbf{z}, \epsilon)) \tag{6}$$

By playing a minimax game between $G_x$ and $D_x$, this adversarial learning is expected to induce a generator $G_x^*$ that can generate samples with the same distribution as the observed training instances. Together with the generation loss in Eq.(5), we expect the mapping relation from label vectors to samples induced by $G_x^*$ can be consistent with the observed data. Moreover, the consistency of the mapping relation induced by $G_x$ and the inverse mapping from samples to label vectors through the prediction network $F$ can be further strengthened by enforcing an auxiliary classification loss on the generated data:

$$\mathcal{L}_{c'}(F, G_x) = \mathbb{E}_{\substack{\mathbf{z} \sim P_\mathbf{z} \\ \epsilon \sim P_\epsilon}} \ell_{c'}\big(F(G_x(\mathbf{z}, \epsilon)), \mathbf{z}\big) \tag{7}$$

where $\ell_{c'}(\cdot, \cdot)$ can be a cross-entropy loss between the label prediction probability vector and the sampled true label indicator vector.

### 3.4 LEARNING THE MGPLL MODEL

By integrating the classification loss in Eq.(4), the adversarial losses in Eq.(2) and Eq.(6), the generation loss in Eq.(5) and the auxiliary classification loss in Eq.(7) together, MGPLL learning can be

Table 1: Win/tie/loss counts of pairwise t-test (at 0.05 significance level) between MGPLL and each comparison method.

| | MGPLL vs – | | | | |
|---|---|---|---|---|---|
| | SURE | PALOC | CLPL | PL-SVM | PL-KNN |
| varying $p$ $[r=1]$ | 25/14/3 | 32/10/0 | 36/6/0 | 37/5/0 | 37/5/0 |
| varying $p$ $[r=2]$ | 27/13/2 | 33/9/0 | 33/9/0 | 38/4/0 | 35/7/0 |
| varying $p$ $[r=3]$ | 26/14/2 | 32/10/0 | 32/10/0 | 36/6/0 | 34/8/0 |
| varying $\epsilon$ $[p, r=1]$ | 25/17/0 | 30/12/0 | 32/10/0 | 35/7/0 | 33/9/0 |
| Total | **103/58/7** | **127/41/0** | **133/35/0** | **146/22/0** | **139/29/0** |

formulated as the following min-max optimization problem:

$$\min_{G_n, G_x, F} \max_{D_n, D_x} \mathcal{L}_c(F, G_n) + \mathcal{L}_{adv}^n(G_n, D_n) + \alpha \mathcal{L}_{adv}^x(G_x, D_x) + \beta \mathcal{L}_g(F, G_n, G_x) + \gamma \mathcal{L}_{c'}(F, G_x) \tag{8}$$

where $\alpha$, $\beta$ and $\gamma$ are trade-off hyperparameters. The learning of the overall model corresponds to a minimax adversarial game. We develop a batch-based stochastic gradient descent algorithm to solve it by conducting minimization over $\{G_n, G_x, F\}$ and maximization over $\{D_n, D_x\}$ alternatively. The overall training algorithm is provided in the appendix.

## 4 EXPERIMENT

We conducted extensive experiments on both controlled synthetic PL datasets and real-world PL datasets to investigate the empirical performance of the proposed model. In this section, we present our experimental settings, comparison results and discussions.

### 4.1 EXPERIMENT SETTING

**Datasets** The synthetic datasets are generated from six UCI datasets, *ecoli, deter, vehicle, segment, satimage and letter*. From each UCI dataset, we generated synthetic PL datasets using three controlling parameters $p, r$ and $\epsilon$, following the controlling protocol in previous studies (Wu & Zhang, 2018; Xu et al., 2019a; Lei & An, 2019). Among the three parameters, $p$ controls the proportion of instances that have noise candidate labels, $r$ controls the number of false positive labels, and $\epsilon$ controls the probability of a specific false positive label co-occurring with the true label. Under different parameter configurations, multiple PL variants can be generated from each UCI dataset. Given that both random noise labels and target label-dependent noise labels may exist in real-world applications, we considered two types of settings. In the first type of setting, we consider random noise labels with the following three groups of configurations: (I) $r = 1$, $p \in \{0.1, 0.2, \cdots, 0.7\}$; (II) $r = 2$, $p \in \{0.1, 0.2, \cdots, 0.7\}$; and (III) $r = 3$, $p \in \{0.1, 0.2, \cdots, 0.7\}$. In the second type of setting, we consider the target label-dependent noise labels with the following configuration: (IV) $p = 1, r = 1, \epsilon \in \{0.1, 0.2, \cdots, 0.7\}$. In total, the four groups of configurations provide us 168 (28 configurations $\times$ 6 UCI datasets) synthetic PL datasets.

We used five real-world PL datasets that are collected from several application domains, including FG-NET (Panis & Lanitis, 2014) for facial age estimation, Lost (Cour et al., 2011), Yahoo! News (Guillaumin et al., 2010) for automatic face naming in images or videos, MSRCv2 (Dietterich & Bakiri, 1994) for object classification, and BirdSong (Briggs et al., 2012) for bird song classification.

**Comparison Methods** We compared the proposed MGPLL approach with the following PL methods, each configured with the suggested parameters according to the respective literature: PL-KNN (Hüllermeier & Beringer, 2006), PL-SVM (Nguyen & Caruana, 2008), CLPL (Cour et al., 2011), PALOC (Wu & Zhang, 2018), and SURE (Lei & An, 2019).

### 4.2 RESULTS ON SYNTHETIC PL DATASETS

We conducted experiments on two types of synthetic PL datasets generated from the UCI datasets, with random noise labels and target label-dependent noise labels, respectively. For each PL dataset,

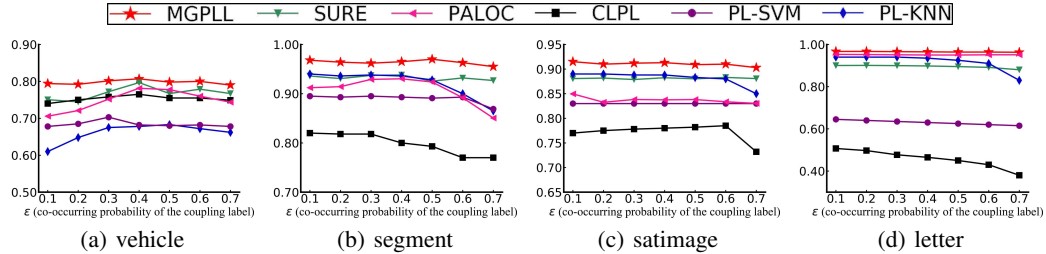

Figure 2: Test accuracy of each comparison method as $\epsilon$ increases from 0.1 to 0.7 (with 100% partially labeled examples $[p = 1]$ and one false positive candidate label $[r = 1]$).

Table 2: Test accuracy (mean±std) of each comparison method on the real-world PL datasets. ●/○ indicates whether MGPLL is statistically superior/inferior to the comparison method on each dataset (pairwise t-test at 0.05 significance level).

| | MGPLL | SURE | PALOC | CLPL | PL-SVM | PL-KNN |
|---|---|---|---|---|---|---|
| FG-NET | $0.079_{\pm0.024}$ | $0.068_{\pm0.032}$ | $0.064_{\pm0.019}$ | $0.063_{\pm0.027}$ | $0.063_{\pm0.029}$ | $0.038_{\pm0.025}$● |
| FG-NET(MAE3) | $0.468_{\pm0.027}$ | $0.458_{\pm0.024}$ | $0.435_{\pm0.018}$● | $0.458_{\pm0.022}$ | $0.356_{\pm0.022}$● | $0.269_{\pm0.045}$● |
| FG-NET(MAE5) | $0.626_{\pm0.022}$ | $0.615_{\pm0.019}$ | $0.609_{\pm0.043}$● | $0.596_{\pm0.017}$● | $0.479_{\pm0.016}$● | $0.438_{\pm0.053}$● |
| Lost | $0.798_{\pm0.033}$ | $0.780_{\pm0.036}$● | $0.629_{\pm0.056}$ | $0.742_{\pm0.038}$● | $0.729_{\pm0.042}$● | $0.424_{\pm0.036}$● |
| MSRCv2 | $0.533_{\pm0.021}$ | $0.481_{\pm0.036}$● | $0.479_{\pm0.042}$● | $0.413_{\pm0.041}$● | $0.461_{\pm0.046}$● | $0.448_{\pm0.037}$● |
| BirdSong | $0.748_{\pm0.020}$ | $0.728_{\pm0.024}$● | $0.711_{\pm0.016}$● | $0.632_{\pm0.019}$● | $0.660_{\pm0.037}$● | $0.614_{\pm0.021}$● |
| Yahoo! News | $0.678_{\pm0.008}$ | $0.644_{\pm0.015}$● | $0.625_{\pm0.005}$● | $0.462_{\pm0.009}$● | $0.629_{\pm0.010}$● | $0.457_{\pm0.004}$● |

ten-fold cross-validation is performed and the average test accuracy results are recorded. Figure 2 presents the comparison results for the configuration setting (IV) on four datasets. We can see that the proposed MGPLL consistently outperforms all the other methods.

To statistically study the significance of the performance gains achieved by MGPLL over the other comparison methods, we conducted pairwise t-test at 0.05 significance level based on the comparison results of ten-fold cross-validation over all the 168 synthetic PL datasets obtained from all the different configuration settings. The detailed win/tie/loss counts between MGPLL and each comparison method are reported in Table 1. From the results, we have the following observations: (1) MGPLL achieves superior or at least comparable performance against PALOC, CLPL, PL-SVM and PL-KNN in all cases, which is not easy given the comparison methods have different strengths across different datasets. (2) MGPLL significantly outperforms PALOC, CLPL, PL-SVM and PL-KNN in 75.6%, 79.1%, 86.9% and 82.7% of the cases respectively, and produces ties in the remaining cases. (3) MGPLL significantly outperforms SURE in 61.3% of the cases, achieves comparable performance with SURE in 34.5% of the cases, while being outperformed by SURE in only 4.2% of the cases. (4) On the PL datasets with target label-dependent noise labels, we can see that MGPLL significantly outperforms SURE , PALOC, CLPL, PL-SVM, PL-KNN in 59.5%, 71.4%, 76.2%, 83.3%, 78.6% of the cases respectively. (5) It is worth noting that MGPLL is never significantly outperformed by any comparison method on datasets with label-dependent noise labels. In summary, these results on the controlled PL datasets clearly demonstrate the effectiveness of MGPLL for partial label learning under different settings.

## 4.3 RESULTS ON REAL-WORLD PL DATASETS

We compared the proposed MGPLL method with the comparison methods on five real-world PL datasets. For each dataset, ten-fold cross-validation is conducted. The mean test accuracy and the standard deviation results are reported in Table 2. Moreover, statistical pairwise t-test at 0.05 significance level is conducted to compare MGPLL with each comparison method based on the results of ten-fold cross-validation. The significance results are indicated in Table 2 as well. Note that the average number of candidate labels (avg.#CLs) of FG-NET dataset is quite large, which causes poor performance for all the comparison methods. For better evaluation of this facial age estimation task, we employ the conventional mean absolute error (MAE) (Zhang et al., 2016) to conduct two extra experiments. Two extra test accuracies are reported on the FG-NET dataset where

Table 3: Comparison results of MGPLL and its five ablation variants.

| | MGPLL | CLS-w/o-advn | CLS-w/o-advx | CLS-w/o-g | CLS-w/o-aux | CLS |
|---|---|---|---|---|---|---|
| FG-NET | $0.079_{\pm0.024}$ | $0.061_{\pm0.024}$ | $0.072_{\pm0.020}$ | $0.068_{\pm0.029}$ | $0.076_{\pm0.022}$ | $0.057_{\pm0.016}$ |
| FG-NET(MAE3) | $0.468\pm0.027$ | $0.430_{\pm0.029}$ | $0.451_{\pm0.032}$ | $0.436_{\pm0.038}$ | $0.456_{\pm0.033}$ | $0.420_{\pm0.420}$ |
| FG-NET(MAE5) | $0.626_{\pm0.022}$ | $0.583_{\pm0.055}$ | $0.605_{\pm0.031}$ | $0.590_{\pm0.045}$ | $0.612_{\pm0.044}$ | $0.570_{\pm0.034}$ |
| Lost | $0.798_{\pm0.033}$ | $0.623_{\pm0.037}$ | $0.754_{\pm0.032}$ | $0.687_{\pm0.026}$ | $0.782_{\pm0.043}$ | $0.609_{\pm0.040}$ |
| MSRCv2 | $0.533_{\pm0.021}$ | $0.472_{\pm0.030}$ | $0.480_{\pm0.038}$ | $0.497_{\pm0.031}$ | $0.526_{\pm0.036}$ | $0.450_{\pm0.037}$ |
| BirdSong | $0.748_{\pm0.020}$ | $0.728_{\pm0.010}$ | $0.732_{\pm0.011}$ | $0.716_{\pm0.011}$ | $0.742_{\pm0.024}$ | $0.674_{\pm0.016}$ |
| Yahoo! News | $0.678_{\pm0.008}$ | $0.645_{\pm0.008}$ | $0.675_{\pm0.009}$ | $0.648_{\pm0.014}$ | $0.671_{\pm0.012}$ | $0.610_{\pm0.015}$ |

a test sample is considered to be correctly predicted if the difference between the predicted age and the ground-truth age is less than 3 years (MAE3) or 5 years (MAE5). From Table 2 we have the following observations: (1) Comparing with all the other five PL methods, MGPLL consistently produces the best results on all the datasets, with remarkable performance gains in many cases. For example, MGPLL outperforms the best alternative comparison methods by 5.2%, 3.4% and 2.0% on MSRCv2, Yahoo! News and Birdsong respectively. (2) Out of the total 35 comparison cases (5 comparison methods × 7 datasets), MGPLL significantly outperforms all the comparison methods across 77.1% of the cases, and achieves competitive performance in the remaining 22.9% of cases. (3) It is worth noting that the performance of MGPLL is never significantly inferior to any other comparison method. These results again validate the efficacy of the proposed method.

## 4.4 ABLATION STUDY

The objective function of MGPLL contains five loss terms: classification loss, adversarial loss at the label level, adversarial loss at the feature level, generation loss and auxiliary classification loss. To assess the contribution of each part, we conducted an ablation study by comparing MGPLL with the following ablation variants: (1) CLS-w/o-advn, which drops the adversarial loss at the label level. (2) CLS-w/o-advx, which drops the adversarial loss at the feature level. (3) CLS-w/o-g, which drops the generation loss. (4) CLS-w/o-aux, which drops the auxiliary classification loss. (5) CLS, which only uses the classification loss by dropping all the other loss terms. The comparison results are reported in Table 3. We can see that comparing to the full model, all five variants produce inferior results in general and have performance degradations to different degrees. This demonstrates that the different components in MGPLL all contribute to the proposed model to some extend. From Table 3, we can also see that the variant CLS-w/o-advn has a relatively larger performance degradation by dropping the adversarial loss at the label level, while the variant CLS-w/o-aux has a small performance degradation by dropping the auxiliary classification loss. This makes sense as by dropping the adversarial loss for learning noise label generator, the generator can produce poor predictions and seriously impact the label denoising of the MGPLL model. This suggests that our non-random noise label generation through adversarial learning is a very effective and important component for MGPLL. For CLS-w/o-aux, as we have already got the classification loss on real data, it is reasonable to see that the auxiliary classification loss on generated data can help but is not critical. Overall, the ablation results suggest that the proposed MGPLL is effective.

## 5 CONCLUSION

In this paper, we proposed a novel multi-level generative model, MGPLL, for partial label learning. MGPLL uses a conditional label level generator to model the target label dependent non-random noise label appearances, which directly performs candidate label denoising, while using a conditional feature level generator to generate data samples from denoised label vectors. Moreover, a prediction network is incorporated to predict the denoised true label of each instance from its input features, which forms bi-directional inverse mappings between labels and features, together with the data feature generator. The adversarial learning of the overall model simultaneously identifies true labels of the training instances from both the observed data features and the observed candidate labels, while inducing accurate prediction networks that map input feature vectors to (denoised) true label vectors. We conducted extensive experiments on real-world and synthesized PL datasets. The proposed MGPLL model demonstrates the state-of-the-art PL performance.

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

---

**Algorithm 1** Minibatch stochastic gradient descent.

---

**Input**: $S$: the PL training set; $\alpha, \beta, \gamma$: the trade-off hyperparameters;
.        $c$: the clipping parameter; $m$: minibatch size.

   **for** number of training iterations **do**

      Sample a minibatch $B = \{(\mathbf{x}_i, \mathbf{y}_i)\}_{i=1}^m$ of $m$ samples from $S$.

      Sample $m$ noise values $\{\epsilon_1, \cdots, \epsilon_i, \cdots, \epsilon_m\}$ from a prior $P(\epsilon)$.

      Sample $m$ label vectors $\{\mathbf{z}_1, \cdots, \mathbf{z}_i, \cdots, \mathbf{z}_m\}$ from a prior $P_{\mathbf{z}}$.

      Update $D_n, D_x$ by ascending their stochastic gradients:

$$\nabla_{\Theta_{D_n,D_x}} \frac{1}{m} \sum_{i=1}^m \left\{ \left( D_n(\mathbf{y}_i) - D_n(G_n(\mathbf{z}_i, \epsilon_i) \oplus \mathbf{z}_i) \right) + \alpha \left( D_x(\mathbf{x}_i) - D_x(G_x(\mathbf{z}_i, \epsilon_i)) \right) \right\}$$

      Perform WGAN adjustment: $\Theta_{D_\mathbf{n}, D_\mathbf{x}} \leftarrow \mathrm{clip}(\Theta_{D_\mathbf{n}, D_\mathbf{x}}, -c, c)$

      Sample $m$ noise values $\{\bar{\epsilon}_1, \cdots, \bar{\epsilon}_i, \cdots, \bar{\epsilon}_m\}$ from a prior $P(\epsilon)$.

      Update $G_\mathbf{n}, G_\mathbf{x}, F$ by stochastic gradient descent:

$$\nabla_{\Theta_{G_n,G_x,F}} \frac{1}{m} \sum_{i=1}^m \left\{ \begin{array}{l} \ell_c(F(\mathbf{x}_i), \mathbf{y}_i \ominus G_n(F(\mathbf{x}_i), \epsilon_i)) - D_n(G_n(\mathbf{z}_i, \epsilon_i) \oplus \mathbf{z}_i) - \alpha D_x(G_x(\mathbf{z}_i, \bar{\epsilon}_i)) + \\ \beta \ell_g \left( G_x(\mathbf{y}_i \ominus G_n(F(\mathbf{x}_i), \epsilon_i), \bar{\epsilon}_i), \mathbf{x}_i \right) + \gamma \ell_{c'} F(G_x(\mathbf{z}_i, \bar{\epsilon}_i), \mathbf{z}_i) \end{array} \right\}$$

   **end for**

---

Table 4: Characteristics of the UCI datasets (left side) and the real-world PL datasets (right side).

| Dataset | #Example | #Feature | #Class | Dataset | #Example | #Feature | #Class | avg.#CLs |
|---|---|---|---|---|---|---|---|---|
| ecoli | 336 | 7 | 8 | FG-NET | 1,002 | 262 | 78 | 7.48 |
| deter | 358 | 23 | 6 | Lost | 1,122 | 108 | 16 | 2.23 |
| vehicle | 846 | 18 | 4 | MSRCv2 | 1,758 | 48 | 23 | 3.16 |
| segment | 2310 | 18 | 7 | BirdSong | 4,998 | 38 | 13 | 2.18 |
| satimage | 6,345 | 36 | 7 | Yahoo! News | 22,991 | 163 | 219 | 1.91 |
| letter | 20,000 | 16 | 26 | | | | | |

## A  APPENDIX

### A.1  THE OVERALL TRAINING ALGORITHM

The overall training algorithm for solving the formulated min-max optimization problem in Eq.(8) is outlined in Algorithm 1.

### A.2  THE CHARACTERISTICS OF THE DATASETS

The characteristics of the UCI datasets and the real-world PL datasets are summaized in Table 4.

### A.3  IMPLEMENTATION DETAILS

The proposed MGPLL model has five component networks, all of which are designed as multilayer perceptrons with Leaky ReLu activation for the middle layers. The noise label generator is a four-layer network with sigmoid activation in the output layer. The conditional data generator is a five-layer network with tanh activation in the output layer, while batch normalization is deployed in its three middle layers. The predictor is a three-layer network with softmax activation in the output layer. Both the noise label discriminator and the data discriminator are three-layer networks without activation in the output layer. We used the RMSProp (Tieleman & Hinton, 2012) optimizer in our implementation and the mini-batch size $m$ is set to 32. We selected the hyperparameters $\alpha$, $\beta$ and $\gamma$ from {0.001, 0.01, 0.1, 1, 10} in a heuristic way based on the classification loss value $\mathcal{L}_c$ in the training objective function; that is, we chose their values that lead to the smallest training $\mathcal{L}_c$ loss.

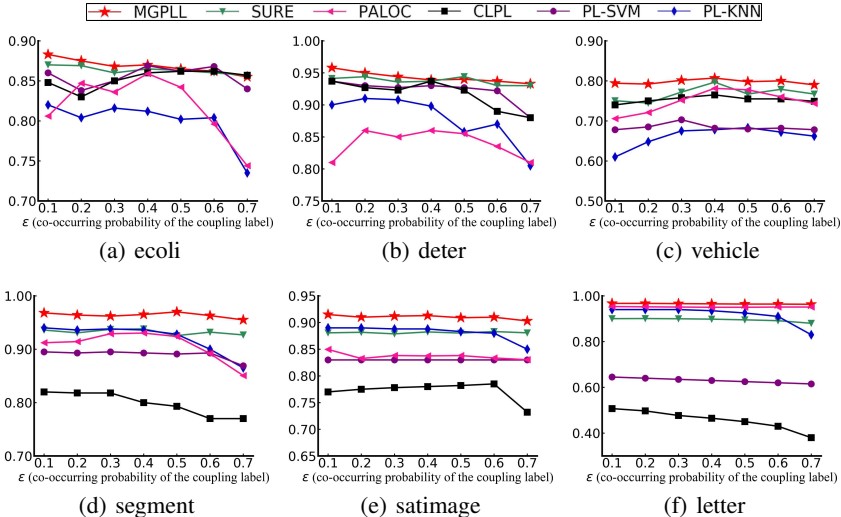

Figure 3: Test accuracy of each comparison method changes as $\epsilon$ (co-occurring probability of the coupling label) increases from 0.1 to 0.7 (with 100% partially labeled examples $[p = 1]$ and one false positive candidate label $[r = 1]$).

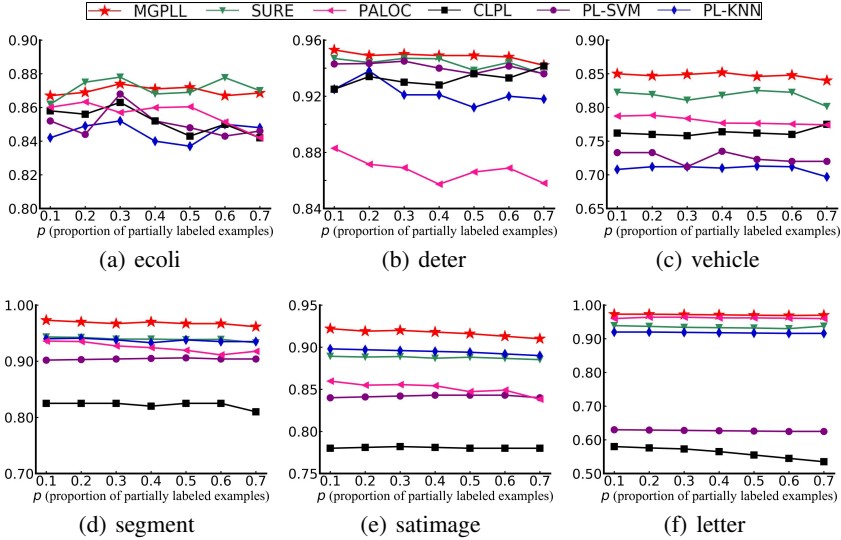

Figure 4: Test accuracy of each comparison method changes as $p$ (proportion of partially labeled examples) increases from 0.1 to 0.7 (with one false positive candidate label $[r = 1]$).

## A.4 MORE RESULTS ON SYNTHETIC PL DATASETS

We conducted experiments on two types of synthetic PL datasets generated from the UCI datasets, with random noise labels and target label-dependent noise labels, respectively. For each PL dataset, ten-fold cross-validation is performed and the average test accuracy results are recorded. First we study the comparison results over the synthetic PL datasets with target label-dependent noise labels under the PL configuration setting (IV). In this setting, a specific label is selected as the coupled label that co-occurs with the ground-truth label with probability $\epsilon$, and any other label can be randomly chosen as a noisy label with probability $1 - \epsilon$. Figure 3 presents the comparison results for the configuration setting (IV), where $\epsilon$ increases from 0.1 to 0.7 with $p = 1$ and $r = 1$. From Figure 3 we can see that the proposed MGPLL produces impressive results. It consistently outperforms

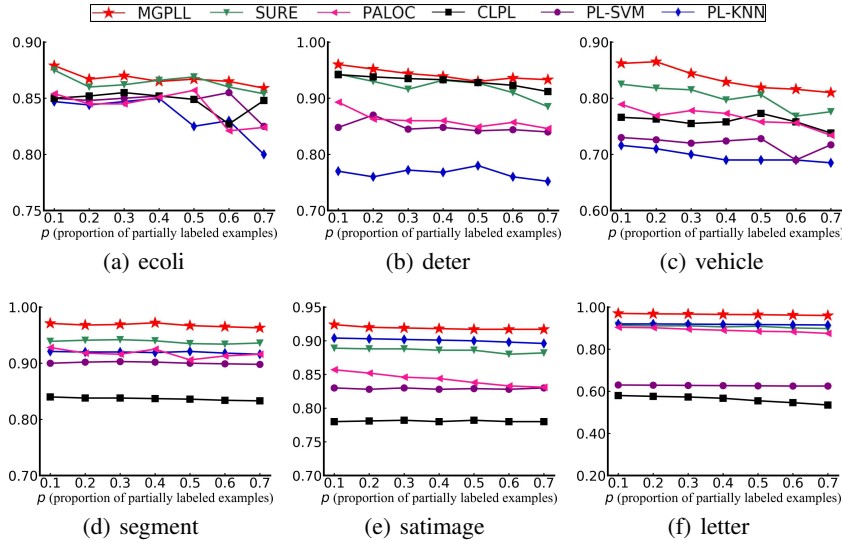

Figure 5: Test accuracy of each comparison method changes as $p$ (proportion of partially labeled examples) increases from 0.1 to 0.7 (with two false positive candidate label $[r = 2]$).

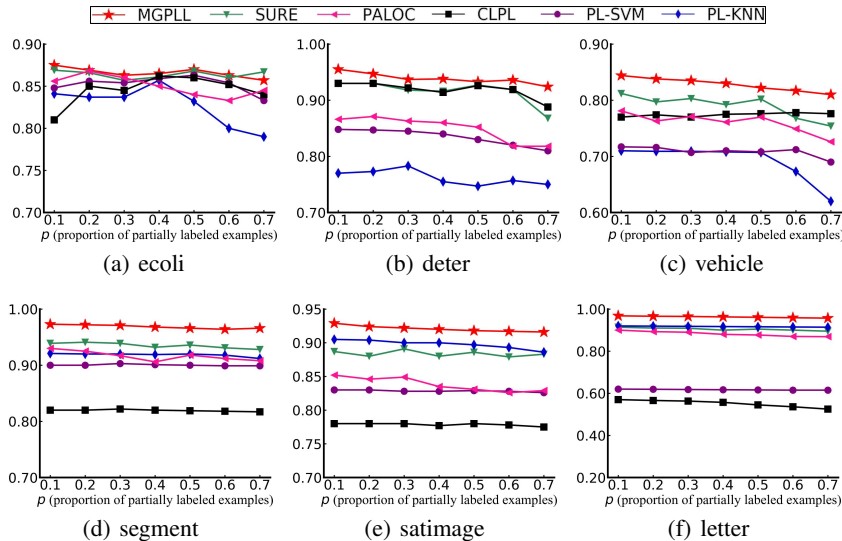

Figure 6: Test accuracy of each comparison method changes as $p$ (proportion of partially labeled examples) increases from 0.1 to 0.7 (with three false positive candidate label $[r = 3]$).

all the other methods across different $\epsilon$ values on four datasets, *vehicle, segment, satimage* and *letter*, while achieving remarkable performance gains on *segment* and *satimage*. On the other two datasets, *ecoli* and *deter*, MGPLL also produces the best results in most cases and remains to be the most effective method. By contrast, the performance of the other comparison methods varies largely across different datasets. For example, CLPL and SURE demonstrate good performance on *ecoli, deter* and *vehicle*, but presents inferior results than PL-KNN in many cases of the other three datasets. PALOC and PL-SVM have the same drawback of producing poor results on some datasets. Our proposed MGPLL demonstrates good overall performance across these varying cases.

We also conducted experiments on the PL datasets with random noise labels produced under the PL configuration settings (I), (II) and (III). The comparison results in these three sets of configurations are reported in Figure 4, Figure 5 and Figure 6 respectively. From these figures we can see

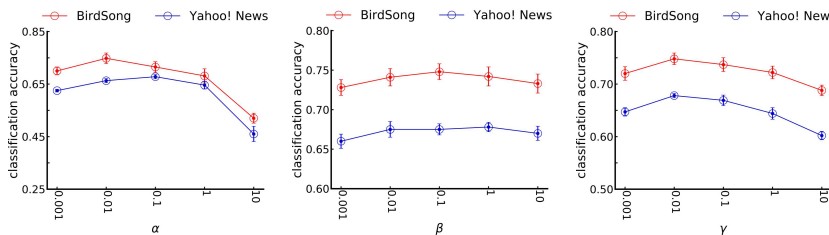

Figure 7: Parameter sensitivity analysis for MGPLL on the Lost and MSRCv2 datasets.

that the proposed MGPLL (with noise label generator $G_n(\epsilon)$) achieves similar positive comparison results as in the configuration setting (IV). In particular, the proposed method achieves remarkable performance gains on four of the overall six datasets, *segment, satimage, vehicle* and *letter*.

## A.5 PARAMETER SENSITIVITY ANALYSIS

We also conducted parameter sensitivity analysis on two real-world PL datasets BirdSong and Yahoo! News datasets to study how the trade-off hyperparameters $\alpha, \beta$ and $\gamma$ influence the performance of MGPLL. We conducted the experiments by using different combination settings of the $\alpha, \eta$ and $\gamma$ values from $\{0.001, 0.01, 0.1, 1, 10\}$. We vary each parameter's value by keeping the other two fixed at their best setting. Note that a larger value for $\alpha, \beta$ and $\gamma$ will provide larger weight to the feature level WGAN loss, generation loss and auxiliary classification loss respectively.

The three figures in Figure 7 report the average test results as well as standard deviations for different $\alpha, \beta$ and $\gamma$ values respectively. We can see that when $\alpha$ is very small, the performance of MGPLL is not very good since the feature level WGAN loss is not allowed to contribute much to the learning. With the increase of $\alpha$, the performance improves, which suggests that the WGAN loss is important. When $\alpha$ is too large, the performance degrades as the WGAN loss dominates. This is reasonable since the WGAN loss is expected to help the predictive model, rather than dominate the learning process. A similar phenomenon can be observed for $\gamma$. For the parameter $\beta$, the proposed method performs bad when $\beta$ is very small. With the increase of $\beta$, the performance of MGPLL improves and remains relatively stable in a broader range, i.e., $\beta \in [0.01, 1]$. It shows that the proposed model is not very sensitivity to the $\beta$ parameter within the considered range of values.

