# OpenReview forum: "Multi-Level Generative Models for Partial Label Learning with Non-random Label Noise"
_ICLR.cc/2021/Conference — Reject_

### Official Review · AnonReviewer4 · 2020-10-24
**Novel idea with good performance**

**Rating:** 7
**Confidence:** 4

**Review:**

Overall, I like the idea of this paper and it is well-written. In this paper, the authors propose multi-level generative models for partial label learning with non-random label noise. It consists of five components: the conditional noise label generator which models the noise labels conditioning on the ground-truth label at the label level; the conditional data generator which generates data samples at the feature level conditioning on the denoised label vectors; the discriminator which separates the generated candidate label vectors from the observed candidate label vectors in the real training data; the discriminator which separates the generated samples from the real data in the feature space; and the prediction network which predicts the denoised label for each sample from its input features. With the proposed minmax adversarial loss, the proposed framework achieved state-of-the-art performance for partial label learning.\
\
I think this paper has the following advantages:
1. It is novel to exploit multi-level generative models to model non-random noise labels for the partial label learning problem.
2. The experiments in this paper are complete and thorough. The authors have tested the model in many datasets and designed the ablation study to verify the effect of each loss.
3. The proposed model achieved the state-of-art results.

Despite the above advantages, I still have the following questions:
1. Is the true label vector $\mathbf{z}$ given for each training sample as the ground truth or it is sampled from the multinomial distribution?
2. In Algorithm 1, why the parameter $\Theta$ needs to be limited in $[-c, c]$?
3. In equation 3, it is not clear what the “$n$” in $\mathbf{y}_n$ stands for? In the previous context, $n$ is the size of the training set.
4. In Fig. 1, the input for $\mathcal{L}_c$ is $\mathbf{z}$ and $F(\mathbf{x})$, while in equation 4 it is $F(\mathbf{x})$ and $\mathbf{y}\ominus G_n(F(\mathbf{x}),\epsilon))$

---

### Official Review · AnonReviewer2 · 2020-10-29
**containing some interesting ideas while not so solid**

**Rating:** 6
**Confidence:** 3

**Review:**

This paper presents a multi-level generative model for partial label learning. The basic idea is to use a conditional noise label generation network to model the label noise. The noise label generator and the data feature generator are learned in an adversarial manner. Experiments on synthesized and real-world data sets show competitive performance.

Overall the idea modeling conditional noise label is interesting. The paper is easy to follow. My concerns are mainly as following.

1.	It is not unclear for me for the motivation that previous methods are based on the separate label distribution estimation steps or the error-prone label confidence estimation process. Such kinds of approaches do not mean that they are not good methods. Actually using the separate label distribution estimation steps or the error-prone label confidence estimation process may be benefit to be more accurate and more efficient. Therefore, such motivation is not so convincing, and leads to an important new contribution.
2.	The second motivation is based on non-random noise. This is an interesting observation. The proposed method involves a conditional probability to model the correlation. However, this might be sensitive and restrict. In practical, such kind of information may be not correct for infrequent patterns. For frequent patterns, this could be trivial to hold.

---

### Official Review · AnonReviewer1 · 2020-10-30
**Strong experimental results but weak motivation in model design**

**Rating:** 5
**Confidence:** 4

**Review:**

This submission proposes a new method of learning from data with partially observed labels. In this problem, every instance has a label candidate set, which contains the true label. This submission introduces adversarial learning to improve the disambiguation of inexact labels. Particularly, there are two adversarial learning component. In the first component, a generator tries to match the distribution of label candidate sets given the "true" label of an instance. In the second component, a generator tries to learn the distribution of instances give their "true" labels. Since the the "true" label is not accessible, the "true" label is actually from a predictive model.

The submission has done extensive experiments and shows that the proposed method outperforms several baseline methods.

In a summary, the experiment results of this submission is strong, but I feel the motivation of the model design is not clear.

My biggest question is the motivation behind adversarial training. To make it simple, adversarial training aims to match a generative distribution to the data distribution.

1. Loss (4) is somewhat reasonable to me. Can I interpret it as: F(x) predict z, and z generate a label vector y'; if y' does not match the partial label y, then there is a loss? Then the model needs to learn both the predictive distribution p(z | x) through F, and the label distribution p(y | z).

If my interpretation is correct, have you considered to use alternative distributions for p(y | z) instead of a generator? There are many choices such as Restricted Boltzmann machine. Since you treat y as continuous variables, there are even more choices such as a decoder in a conditional variational autoencoder.  I don't mean to say that GAN distribution is not a good choice, but if you could make this task clear so people may consider alternative choices.

2.The generative distribution p(x | z) is more confusing. The structure in (5) seems like an encoder-decoder: decoding x, y gives z, and z should recover x. With this loss, the model has some flavor of generative modeling because it models the feature vector x. As a principle in machine learning, generative modeling is less powerful than discriminative models in classification tasks, so I don't see why we want to model input features.

3. In the ablation study, have you tried to use only the classification model and losses from adversarial labels?

4. Since the model has better performance than baseline methods, there might be several explanations.

1) differences in base models. What are base predictive models of F in your algorithm and equivalent in competing methods? Is the difference significant? In the results from synthetic datasets, p(y | z) does not contain much information since noise labels are randomly sampled. The proposed method still outperform baseline methods by a good margin. Is this an evidence that the model is still better without much contribution from the label generator? In another word, if you replace G_n(z, \epsilon) by your groundtruth distribution, the model should perform even better?

2) The combination of the generative model Gx(Z, epsilon), which works like a generative model. When it works with F(x) and other loss terms, the entire model is like a combination of two neural networks. This might provide extra classification power?

5. The loss in (4) is unnatural to me: the loss is computed from probabilities, and square loss may not be the best choice.

I am asking a lot of questions, trying to explain why the experiment results. In another work, I wish the submission could answer some of these questions.

The proposed model also have many parameters and neural network components. I don't how easy it is for others to tune the model.

The writing of the submission also has many issues. A lot of symbols are not well defined. A few examples.
1. n has two meaning: number of instances and "noisy" labels.
2. what is p_z?
3. The minimization problem in 2 is invalid since the true label z is unknown.

---

### Decision · Program_Chairs · 2021-01-07
**Final Decision**

**Decision:**

Reject

**Comment:**

Dear Authors,

Thank you very much for your detailed feedback to the reviewers in the rebuttal phase. This certainly clarified some of the concerns raised by the reviewers and contributed highly to deepen their understanding of your work.

We positively evaluated the novelty and the superior empirical performance of the proposed method. However, we still have concern about the justification since the proposed model is so complex that it is not clear what was the key for the good performance.

For this reason, I suggest rejection of this submission, in comparison with many other strong submissions. I hope that the reviewers' feedback is useful for improving your work for future publication.

Best,
AC